# Metal-Free Eliminative C-H Arylthiolation of 2*H*-Imidazole N-Oxides with Thiophenols

Egor A. Nikiforov [1] , Nailya F. Vaskina [1], Timofey D. Moseev [1] , Mikhail V. Varaksin [1,2,*] ,
Valery N. Charushin [1,2] and Oleg N. Chupakhin [1,2]

[1]  Institute of Chemical Engineering, Ural Federal University, 19 Mira Street, 620002 Ekaterinburg, Russia;
     chupakhin@ios.uran.ru (O.N.C.)
[2]  I.Ya. Postovsky Institute of Organic Synthesis, Ural Branch of the Russian Academy of Sciences, 22 S.
     Kovalevskaya Street, 620990 Ekaterinburg, Russia
[*]  Correspondence: m.v.varaksin@urfu.ru

**Abstract:** A synthetic strategy based on reactions of cyclic imine oxides, namely 2*H*-imidazole 1-oxides, with thiophenols mediated by acetyl chloride was successfully applied as a convenient tool to obtain a series of novel azaheterocyclic molecules, including water-soluble hydrochloride forms. Optimized reaction conditions found herein for the nucleophilic substitution of hydrogen ($S_N^H$) in non-aromatic azaheterocyclic substrates via the "addition-elimination" ($S_N^H$ AE) scheme enabled 15 arylthiolated 2*H*-imidazoles to be prepared in yields of up to 90%. The developed methodology discloses an original synthetic way to obtain numerous azaheterocyclic molecules, which are of interest in the field of medicinal chemistry and materials science.

**Keywords:** azaheterocycles; imidazoles; thiophenols; C-H functionalization; nucleophilic substitution of hydrogen





## 1. Introduction

The arylthiol and imidazole moieties are known to be key structural motifs of organic compounds with various types of pharmaceutical activities and functional materials [1–5]. In particular, the molecules bearing imidazole rings also provide a huge number of biological activities and are used as active pharmaceutical ingredients in many drugs. For example, Losartan is an angiotensin II receptor agonist that is used as an antihypertensive agent [6]. The imidazole-containing compounds also demonstrate antifungal activity; for instance, clotrimazole is used successfully for treating systemic Candida infections, pseudallescheriasis, and some refractory cases of cryptococcal meningitis [7]. Besides, im-idazole substrates are also used as antiparasitic drugs (tinidazole, secnidazole, etc.) (Figure 1, top) [8,9]. Moreover, there are known imidazole-based structures characterized by anticancer effects as well. Dacarbazine is used for the treatment of metastatic malignant melanoma, Hodgkin lymphoma, sarcoma, and islet cell pancreatic carcinoma [10]. Therefore, the elaboration of new ways and approaches for the design and synthesis of azaheterocyclic systems, especially with imidazole moiety, appears to be a key task for modern organic and medicinal chemistry.

Among sulfenyl-derived compounds, the molecules that contain the C-S bond linking the arylthio and the azaheterocyclic moieties are of increasing interest. Substances of this type have found several applications as pharmaceutically active compounds (Figure 1, bottom), particularly as anti-tuberculosis (**I**) and anti-HIV (**II**) agents [11,12]. In addition, benzyl-modified imidazole **III** possesses inhibitory activity regarding the biological target STAT3 associated with the pathogenesis of oncological diseases, namely breast cancer [13]. Also, arylthioindole derivative **IV** is a tubulin inhibitor related to oncogenesis progression [14]. Azathioprine **V** is used for the treatment of rheumatoid arthritis, granulomatosis with polyangiitis, and other diseases [15].

**Figure 1.** Active pharmaceutical ingredients (APIs) based on imidazole (**top**) and azaheterocyclic sulfenylated derivatives (**bottom**: red color indicates the azaheterocyclic fragment; blue color shows the arylthio moiety).

Currently, a limited number of synthetic methods to obtain arylthio(hetero)cycles, particularly imidazoles, have been reported [16–23]. Arylation of thiol or thion group in the imidazole moiety by the reaction with aryl halides is commonly used (Scheme 1a) [13,24]. There are also C-I/S-H couplings of imidazole halides with thiophenols or disulfides (Scheme 1b) [25]. Previously, our group reported the transitional metal-free C-H/C-H coupling reactions of 2*H*-imidazole 1-oxides with various nucleophiles (pyrroles, indoles, and phenols) [26,27]. The latter reactions were developed according to the basic principles of green chemistry, particularly using nontoxic solvents, reducing the number of formed by-products, etc. [28–30]. Following the green chemistry-oriented C-H functionalization synthetic strategy [31–33], namely, reactions of nucleophilic substitution of hydrogen ($S_N^H$) can be successfully used to modify various organic substrates [34]. At the same time, the application of this approach to the direct modification of heterocyclic substrates by S-nucleophiles has not been thoroughly studied yet.

This work deals with the first systematic study of eliminative arylthiolation of 2*H*-imidazole N-oxides by coupling with thiophenols. Furthermore, the reaction conditions optimization, scope, and limitation for the developed method are highlighted (Scheme 1c).

*Arylation of thione and thiols*

**Scheme 1.** Synthetic strategies towards arylthiolated-imidazoles: (**a**) arylation of thiole or thione; (**b**) direct C-I/S-H coupling method; (**c**) the present work (red color indicates the imidazole ring; blue color shows the arylthio moiety).

## 2. Materials and Methods

### 2.1. Experimental Procedure

Nuclear magnetic resonance (NMR) spectra were recorded on the Bruker Avance II (400 MHz) and Bruker Avance II (600 MHz) spectrometers. All $^1$H NMR experiments were reported in δ units, parts per million (ppm), and were measured relative to residual chloroform CDCl$_3$ (7.26 ppm), DMSO (2.50 ppm), or CF$_3$COOD + CD$_3$COOD (2.04 ppm) signals in the deuterated solvent. All $^{13}$C NMR spectra were reported in parts per million (ppm) relative to CDCl$_3$ (77.16 ppm), DMSO-$d_6$ (39.52 ppm), or CF$_3$COOD + CD$_3$COOD (20.0 ppm), and all spectra were obtained with $^1$H decoupling. All coupling constants *J* were reported in Hertz (Hz). The following abbreviations were used to describe peak splitting patterns (s = singlet, d = doublet, t = triplet, dd = doublet of doublet, m = multiplet, and br s = broadened singlet). Copies of NMR spectra are illustrated in Supplementary Materials. The mass spectra were recorded on a mass spectrometer, SHIMADZU GCMS-QP2010 Ultra, with sample ionization by electron impact (EI). The IR spectra were recorded using a Fourier-transform infrared spectrometer (Bruker Corporation, 40 Manning Rd, Billerica, MA, USA) equipped with a diffuse reflection attachment. The elemental analysis was carried out on a Perkin Elmer Instrument (PerkinElmer, Waltham, MA, USA) equipped with CHN PE 2400 II analyzer. The course of the reactions was monitored by TLC on 0.25 mm silica gel plates (60F 254, MACHEREY-NAGEL Inc., 924 Marcon Blvd, Allentown, PA 18109, USA).

Thiophenol, 4-hydroxythiophenol, 3,5-difluorothiophenol, thiosalicylic acid, 4-methoxythiophenol, 2,6-dichlorothiophenol, toluene, ethyl acetate, acetone, hexachloroacetone, chlorobenzene, dimethyl carbonate, hexane, chloroform, acetyl chloride, trichloroacetyl chloride, benzoyl chloride, chlorotrimethylsilane, trifluoroacetyc anhydride, acetic anhydride, and sodium bicarbonate were purchased and used as received.

Additionally, 2,2-Dimethyl-4-phenyl-2*H*-imidazole 1-oxide [35], 3-phenyl-1,4-diazaspiro[4.5]deca-1,3-diene 1-oxide [36], 4-(4-bromophenyl)-2,2-dimethyl-2*H*-imidazole 1-oxide, 4-(4-bromophenyl)-2-ethyl-2-methyl-2*H*-imidazole 1-oxide [37], 2,2-dimethyl-4-(p-tolyl)-2*H*-imidazole 1-oxide [38], and 2,2-dimethyl-4-(naphthalen-2-yl)-2*H*-imidazole 1-oxide [27] were used as starting materials and were prepared according to the literature procedures.

### 2.2. General Procedure for the Synthesis of Hydrochloride Salts of Sulfenyl-Imidazole Derivatives (*3a-k*)

To a vigorously stirred mixture of 2*H*-imidazole 1-oxide **1a-f** (0.5 mmol) and thiophenolic substrate **2a-d** (0.5 mmol) in dimethyl carbonate (4 mL) at 0 °C, acetyl chloride (0.5 mmol) was added. Subsequently, the resulting mixture was allowed to warm up to room temperature and was stirred continuously for 6 h. Then, the resulting precipitate **3** was filtered off, washed with hexane (10 mL), and dried under air.

2,2-Dimethyl-4-phenyl-5-(phenylthio)-2*H*-imidazole hydrochloride (**3a**). Note: In case of hydrochloride compound, $m/z = m/z$ compound $- m/z$ hydrochloride. Colorless solid. Yield: 0.37 mmol (117 mg, 74%), mp = 98–99 °C. $R_f$ 0.18 (hexane/EtOAc, 6:4). $^1$H NMR (CDCl$_3$): δ 8.26 (d, $J$ = 7.3 Hz, 2H); 7.72 (t, $J$ = 7.5 Hz, 1H); 7.64–7.60 (m, 4H); 7.56–7.50 (m, 3H); 1.78 (s, 6H) ppm. $^{13}$C {$^1$H} NMR (CDCl$_3$, BB mode): δ 165.5; 162.9; 134.8; 134.7; 131.6; 130.6; 130.5; 129.7; 125.6; 124.4; 100.4; 24.2 ppm. IR (DRA): ν 1734, 1625, 1519, 1438, 1361, 1281, 994, 889, 832, 755, 724, 708, 684, 570, 515 cm$^{-1}$. MS calcd: $m/z$ 316 [M]$^+$. Found (EI) $m/z$ 280 [M]$^+$. Elemental analysis calcd for: C$_{17}$H$_{17}$ClN$_2$S: C, 64.44; H, 5.51; N, 8.84. Found: C, 64.42; H, 5.52; N, 8.81.

**2**-Phenyl-3-(phenylthio)-1,4-diazaspiro[4.5]deca-1,3-diene hydrochloride (**3b**). Note: In case of hydrochloride compound, $m/z = m/z$ compound $- m/z$ hydrochloride. Light-yellow solid. Yield: 0.27 mmol (96 mg, 54%), mp = 112–113 °C. $R_f$ 0.14 (hexane/EtOAc, 6:4). $^1$H NMR (CDCl$_3$): δ 8.28 (d, $J$ = 7.7 Hz, 2H); 7.71 (t, $J$ = 7.4 Hz, 1H); 7.63–7.59 (m, 4H); 7.53–7.45 (m, 3H); 2.16–2.04 (m, 2H); 1.98–183 (m, 3H); 1.73–1.58 (m, 5H) ppm. $^{13}$C {$^1$H} NMR (CDCl$_3$, BB mode): δ 162.9; 161.0; 134.5; 131.1; 130.7; 130.3; 129.7; 129.5; 129.1; 128.9; 103.3; 34.9; 24.5; 23.5. ppm. IR (DRA): ν 2935, 1698, 1629, 1519, 1473, 1373, 1300, 1140, 1066, 1001, 850, 817, 747, 686, 581 cm$^{-1}$. MS calcd: $m/z$ 356 [M]$^+$. Found (EI): $m/z$ 320 [M - HCl]$^+$. Elemental analysis calcd for: C$_{20}$H$_{21}$ClN$_2$S: C, 67.31; H, 5.93; N, 7.85. Found: C, 67.28; H, 5.94; N, 7.86.

4-(4-Bromophenyl)-2,2-dimethyl-5-(phenylthio)-2*H*-imidazole hydrochloride (**3c**). Note: In case of hydrochloride compound, $m/z = m/z$ compound $- m/z$ hydrochloride. Light-yellow solid. Yield: 0.26 mmol (103 mg, 52%), mp = 128–129 °C. $R_f$ 0.26 (hexane/EtOAc, 6:4). $^1$H NMR (DMSO-$d_6$): δ 7.82–7.81 (m, 2H); 7.78–7.76 (m, 2H); 7.65–7.63 (m, 2H); 7.48–7.44 (m, 3H); 1.39 (s, 6H) ppm. $^{13}$C {$^1$H} NMR (DMSO-$d_6$, APT mode): δ 161.0 (C); 160.7 (C); 133.6 (CH); 131.9 (CH); 130.4 (CH); 130.1 (C); 129.5 (CH); 129.4 (CH); 128.3 (C); 124.7 (C); 102.40 (C); 24.1 (CH$_3$) ppm. IR (DRA): ν 1755, 1627, 1585, 1514, 1362, 1291, 1183, 1126, 1069, 995, 892, 728, 704, 689, 570, 523 cm$^{-1}$. MS calcd: $m/z$ 394 [M]$^+$; 396 [M + 2]$^+$. Found (EI): $m/z$ 358 [M - HCl]$^+$; 360 [M + 2 - HCl]$^+$. Elemental analysis calcd for: C$_{17}$H$_{16}$BrClN$_2$S: C, 51.60; H, 4.08; N, 7.08. Found: C, 51.64; H, 4.07; N, 7.09.

4-(4-Bromophenyl)-2-ethyl-2-methyl-5-(phenylthio)-2*H*-imidazole hydrochloride (**3d**). Note: In case of hydrochloride compound, $m/z = m/z$ compound $- m/z$ hydrochloride. Light-yellow solid. Yield: 0.29 mmol (108 mg, 58%), mp = 111–112 °C. $R_f$ 0.23 (hexane/EtOAc, 6:4). $^1$H NMR (CDCl$_3$): δ 8.08–8.06 (m, 2H); 7.75–7.73 (m, 2H); 7.62–7.60 (m, 2H); 7.57–7.48 (m, 3H); 2.36–2.22 (m, 2H); 1.73 (s, 3H); 0.72 (t, $J$ = 7.3 Hz, 3H) ppm. $^{13}$C {$^1$H} NMR (CDCl$_3$, BB mode): δ 162.5; 160.8; 134.6; 133.0; 132.2; 131.5; 130.6; 129.5; 129.2; 103.6; 82.6; 31.3, 22.8, 8.1 ppm. IR (DRA): ν 2978, 1828, 1708, 1625, 1584, 1515, 1478, 1280, 1068,

1005, 834, 755, 663, 579, 554 cm$^{-1}$. MS calcd: $m/z$ 408 [M]$^+$; 410 [M + 2]$^+$. Found (EI): $m/z$ 372 [M - HCl]$^+$; 374 [M + 2 - HCl]$^+$. Elemental analysis calcd for: C$_{18}$H$_{18}$BrClN$_2$S: C, 52.76; H, 4.43; N, 6.84. Found: C, 52.75; H, 4.43; N, 6.85.

2,2-Dimethyl-4-(phenylthio)-5-(*p*-tolyl)-2*H*-imidazole hydrochloride (**3e**). Note: In case of hydrochloride compound, $m/z$ = $m/z$ compound − $m/z$ hydrochloride. Colorless solid. Yield: 0.21 mmol (70 mg, 42%), mp = 132–133 °C. $R_f$ 0.15 (hexane/EtOAc, 6:4). $^1$H NMR (CDCl$_3$): δ 8.25 (d, $J$ = 8.0 Hz, 2H); 7.62–7.60 (m, 2H); 7.55–7.47 (m, 3H); 7.43 (d, $J$ = 8.0 Hz, 2H); 2.48 (s, 3H); 1.76 (s, 6H) ppm. $^{13}$C {$^1$H} NMR (CDCl$_3$, BB mode): δ 164.1; 163.0; 147.0; 134.6; 131.1; 131.0; 130.5; 130.3; 125.4; 122.3; 100.0; 24.3; 22.1 ppm. IR (DRA): ν 1825, 1624, 1604, 1520, 1439, 1359, 1183, 1132, 993, 891, 757, 740, 688, 570, 517 cm$^{-1}$. MS calcd: $m/z$ 330 [M]$^+$. Found (EI): $m/z$ 294 [M - HCl]$^+$. Elemental analysis calcd for: C$_{18}$H$_{19}$ClN$_2$S: C, 65.34; H, 5.79; N, 8.47. Found: C, 65.32; H, 5.80; N, 8.45.

2,2-Dimethyl-4-(naphthalen-2-yl)-5-(phenylthio)-2*H*-imidazole hydrochloride (**3f**). Note: In case of hydrochloride compound, $m/z$ = $m/z$ compound − $m/z$ hydrochloride. Light-yellow solid. Yield: 0.275 mmol (101 mg, 55%), mp = 117–118 °C. $R_f$ 0.25 (hexane/EtOAc, 6:4). $^1$H NMR (CDCl$_3$): δ 8.87 (s, 1H); 8.30 (d, $J$ = 8.6 Hz, 1H); 8.06 (d, $J$ = 8.7 Hz, 2H); 7.95–7.92 (m, 1H), 7.71–7.62 (m, 4H); 7.56–7.52 (m, 3H); 1.82 (s, 6H) ppm. $^{13}$C {$^1$H} NMR (CDCl$_3$, BB mode): δ 162.9; 160.8; 136.0; 134.8; 132.5; 131.5; 130.8; 130.6; 129.9; 129.5; 129.2; 128.2; 128.0; 127.9; 125.7; 125.4; 100.4; 24.4 ppm. IR (DRA): ν 2156, 1854, 1617, 1523, 1472, 1438, 1389, 1289, 1170, 1018, 985, 941, 848, 684, 641, 583, 564 cm$^{-1}$. MS calcd: $m/z$ 366 [M]$^+$. Found (EI): $m/z$ 330 [M - HCl]$^+$. Elemental analysis calcd for: C$_{21}$H$_{19}$ClN$_2$S: C, 65.75; H, 5.22; N, 7.64. Found: C, 65.79; H, 5.21; N, 7.62.

4-((2,2-Dimethyl-5-phenyl-2*H*-imidazol-4-yl)thio)phenol hydrochloride (**3g**). Note: In case of hydrochloride compound, $m/z$ = $m/z$ compound − $m/z$ hydrochloride. Bright-yellow solid. Yield: 0.4 mmol (133 mg, 80%), mp = 218–219 °C. $R_f$ 0.28 (hexane/EtOAc, 6:4). $^1$H NMR (DMSO-$d_6$): δ 9.97 (br s, 1H); 8.32–8.30 (m, 2H); 7.67–7.53 (m, 1H); 7.49 (t, $J$ = 7.4 Hz, 2H); 7.27–7.25 (m, 2H); 6.77–6.76 (m, 2H); 1.44 (s, 6H) ppm. $^{13}$C {$^1$H} NMR (DMSO-$d_6$, APT mode): δ 163.9 (C); 160.2 (C); 158.4 (C); 133.1 (CH); 131.4 (CH); 130.7 (C); 128.5 (CH); 127.9 (CH); 125.0 (C); 116.3 (CH); 79.6 (C); 27.0 (CH$_3$) ppm. IR (DRA): ν 3156, 2393, 1598, 1581, 1545, 1494, 1330, 1314, 1277, 1229, 1164, 996, 912, 831, 777, 569, 529 cm$^{-1}$. MS calcd: $m/z$ 332 [M]$^+$. Found (EI): $m/z$ 296 [M - HCl]$^+$. Elemental analysis calcd for: C$_{17}$H$_{17}$ClN$_2$OS: C, 61.35; H, 5.15; N, 8.42. Found: C, 61.38; H, 5.14; N, 8.41.

4-((3-Phenyl-1,4-diazaspiro[4.5]deca-1,3-dien-2-yl)thio)phenol hydrochloride (**3h**). Note: In case of hydrochloride compound, $m/z$ = $m/z$ compound − $m/z$ hydrochloride. Bright-green solid. Yield: 0.435 mmol (162 mg, 87%), mp = 225–226 °C. $R_f$ 0.23 (hexane/EtOAc, 6:4). $^1$H NMR (DMSO-$d_6$): δ 10.25 (br s, 1H); 7.87–7.85 (m, 2H); 7.57–7.53 (m, 3H); 7.27–7.25 (m, 2H); 6.78–6.76 (m, 2H); 1.79–1.47 (m, 10H) ppm. $^{13}$C {$^1$H} NMR (DMSO-$d_6$, BB mode): δ 164.2; 160.3; 158.4; 133.1; 131.3; 130.9; 128.5; 127.9; 125.0; 116.3; 104.0; 34.4; 24.6; 23.7 ppm. IR (DRA): ν 3117, 2951, 2413, 1597, 1548, 1495, 1441, 1341, 1284, 1118, 817, 779, 726, 696, 545 cm$^{-1}$. MS calcd: $m/z$ 372 [M]$^+$. Found (EI): $m/z$ 336 [M - HCl]$^+$. Elemental analysis calcd for: C$_{20}$H$_{21}$ClN$_2$OS: C, 64.42; H, 5.68; N, 7.51. Found: C, 64.42; H, 5.67; N, 7.53.

4-((5-(4-Bromophenyl)-2,2-dimethyl-2*H*-imidazol-4-yl)thio)phenol hydrochloride (**3i**). Note: In case of hydrochloride compound, $m/z$ = $m/z$ compound − $m/z$ hydrochloride. Bright-green solid. Yield: 0.45 mmol (185 mg, 90%), mp = 236–237 °C. $R_f$ 0.3 (hexane/EtOAc, 6:4). $^1$H NMR (DMSO-$d_6$): δ 10.02 (br s, 1H); 7.82–7.80 (m, 2H); 7.77–7.76 (m, 2H); 7.39–7.38 (m, 2H); 6.86–6.85 (m, 2H); 1.37 (s, 6H) ppm. $^{13}$C {$^1$H} NMR (DMSO-$d_6$, APT mode): δ 162.0 (C); 161.0 (C); 158.9 (C); 135.9 (CH); 131.9 (CH); 130.4 (CH); 130.2 (C); 124.6 (C); 116.6 (CH); 116.0 (C); 102.0 (C); 24.1 (CH$_3$) ppm. IR (DRA): ν 3120, 2406, 1584, 1542, 1496, 1400, 1331, 1279, 1212, 1116, 1069, 889, 833, 566 cm$^{-1}$. MS calcd: $m/z$ 410 [M]$^+$; 412 [M + 2]$^+$. Found (EI): $m/z$ 374 [M - HCl]$^+$; 376 [M + 2 - HCl]$^+$. Elemental analysis calcd for: C$_{17}$H$_{16}$BrClN$_2$OS: C, 49.59; H, 3.92; N, 6.80. Found: C, 49.60; H, 3.92; N, 6.81.

2-((2,2-Dimethyl-5-phenyl-2*H*-imidazol-4-yl)thio)benzoic acid hydrochloride (**3j**). Note: In case of hydrochloride compound, $m/z$ = $m/z$ compound − $m/z$ hydrochloride. Colorless solid. Yield: 0.325 mmol (117 mg, 65%), mp = 176–177 °C. $R_f$ 0.34 (hexane/EtOAc, 6:4). Note: the

hydrogen from -COOH group is not revealed in CF$_3$COOD $^1$H NMR (CF$_3$COOD + CD$_3$COOD): δ 8.29–8.27 (m, 1H); 7.92–7.90 (m, 2H); 7.83–7.90 (m, 1H); 7.76–7.74 (m, 2H); 7.67–7.63 (m, 1H); 7.56–7.52 (m, 2H); 1.70 (s, 6H) ppm. $^{13}$C {$^1$H} NMR (CF$_3$COOD + CD$_3$COOD): δ 182.1; 175.9; 171.6; 165.7; 139.7; 137.5; 136.2; 135.6; 134.4; 131.4; 130.8; 127.9; 124.4; 102.0; 24.5 ppm. IR (DRA): ν 2811, 2458, 1714, 1632, 1522, 1456, 1384, 1237, 1180, 1117, 1050, 986, 840, 762, 727, 688, 641, 575, 520 cm$^{-1}$. MS calcd: *m/z* 360 [M]$^+$. Found (EI): *m/z* 324 [M - HCl]$^+$. Elemental analysis calcd for: C$_{18}$H$_{17}$ClN$_2$O$_2$S: C, 59.91; H, 4.75; N, 7.76. Found: C, 59.81; H, 4.75; N, 7.75.

4-((3,5-Difluorophenyl)thio)-2,2-dimethyl-5-phenyl-2*H*-imidazole hydrochloride (**3k**). Note: In case of hydrochloride compound, *m/z* = *m/z* compound − *m/z* hydrochloride. Colorless solid. Yield: 0.26 mmol (92 mg, 52%), mp = 105–106 °C. *R*$_f$ 0.38 (hexane/EtOAc, 6:4). $^1$H NMR (DMSO-*d*$_6$): δ 7.84–7.82 (m, 2H); 7.61–7.55 (m, 3H); 7.54–7.50 (m, 2H); 7.39–7.35 (m, 1H); 1.44 (s, 6H) ppm. $^{13}$C {$^1$H} NMR (DMSO-*d*$_6$, BB mode): δ 162.1 (dd, *J* = 248.5, 13.7 Hz); 161.1; 159.8; 132.4 (t, *J* = 11.0 Hz); 131.0; 130.6; 128.8; 128.2; 116.2 (dd, *J* = 20.6, 3.5 Hz); 105.1 (t, *J* = 26.1 Hz); 102.7; 24.0 ppm. $^{19}$F NMR (DMSO-*d*$_6$): -108.40 (s, 2F) ppm. IR (DRA): ν 3011, 1815, 1593, 1547, 1434, 1331, 1284, 1211, 1166, 1120, 934, 871, 725, 693, 672, 655, 594, 571 cm$^{-1}$. MS calcd: *m/z* 352 [M]$^+$. Found (EI): *m/z* 316 [M - HCl]$^+$. Elemental analysis calcd for: C$_{17}$H$_{15}$ClF$_2$N$_2$S: C, 57.87; H, 4.29; N, 7.94. Found: C, 57.84; H, 4.28; N, 7.96.

*2.3. General Procedure for the Synthesis of Sulfenyl-Imidazole Derivatives (**4a,i**)*

A mixture of the corresponding hydrochloride of **3a** or **3i** (0.3 mmol) and NaHCO$_3$ (0.45 mmol) in chloroform (5 mL) was refluxed for 30 min. Then, the reaction mixture was cooled to room temperature, filtered off, and the precipitate was washed with 5 mL of chloroform. The filtrate was combined and evaporated in vacuo to obtain compounds **4a** or **4h** as solids.

2,2-Dimethyl-4-phenyl-5-(phenylthio)-2*H*-imidazole (**4a**). Gray crystals. Yield: 0.35 mmol (98 mg, 100%), mp = 90–91 °C. *R*$_f$ 0.45 (hexane/EtOAc, 6:4). $^1$H NMR (DMSO-*d*$_6$): δ 7.87–7.85 (m, 2H); 7.65–7.63 (m, 2H); 7.58–7.53 (m, 3H); 7.48–7.42 (m, 3H); 1.39 (s, 6H) ppm. Note: one carbon (C) atom has overlapped with carbon (C) on 128.5 ppm. $^{13}$C {$^1$H} NMR (DMSO-*d*6, APT mode): δ 161.9 (C); 161.0 (C); 133.7 (CH); 130.9 (CH); 129.5 (CH); 129.3 (CH); 128.8 (CH); 128.5 (C); 128.3 (CH); 102.3 (C); 24.2 (CH$_3$) ppm. IR (DRA): ν 3060, 2977, 2930, 1697, 1630, 1605, 1562, 1489, 1437, 1214, 1163, 1105, 1025, 982, 775, 750, 687, 568 cm$^{-1}$. MS calcd: *m/z* 280 [M]$^+$. Found (EI): *m/z* 280 [M]$^+$. Elemental analysis calcd for: C$_{17}$H$_{16}$N$_2$S: C, 72.82; H, 5.75; N, 9.99. Found: C, 72.80; H, 5.75; N, 10.00.

4-((5-(4-Bromophenyl)-2,2-dimethyl-2*H*-imidazol-4-yl)thio)phenol (**4i**). Gray crystals. Yield: 0.35 mmol (131.6 mg, 100%), mp = 229–230 °C. *R*$_f$ 0.4 (hexane/EtOAc, 6:4). $^1$H NMR (DMSO-*d*$_6$): δ 9.70 (br s, 1H); 7.84–7.82 (m, 2H); 7.70–7.68 (m, 2H); 7.33–7.31 (m, 2H); 6.82–6.80 (m, 2H); 1.39 (s, 6H) ppm. $^{13}$C {$^1$H} NMR (DMSO-*d*$_6$, APT mode): δ 161.8 (C); 161.0 (C); 158.9 (C); 135.8 (CH); 131.8 (CH); 130.3 (CH); 130.2 (C); 124.5 (C); 116.5 (CH); 116.1 (C); 102.0 (C); 24.1 (CH$_3$) ppm. IR (DRA): ν 3053, 2982, 2931, 1706, 1599, 1576, 1526, 1429, 1359, 1283, 1218, 1164, 1099, 1067, 1009, 988, 827, 740, 678, 569 cm$^{-1}$. MS calcd: *m/z* 374 [M]$^+$; 376 [M + 2]$^+$. Found (EI): *m/z* 374 [M]$^+$; 376 [M + 2]$^+$. Elemental analysis calcd for: C$_{17}$H$_{15}$BrN$_2$OS: C, 54.41; H, 4.03; N, 7.46. Found: C, 54.40; H, 4.03; N, 7.47.

4-((4-Methoxyphenyl)thio)-2,2-dimethyl-5-phenyl-2*H*-imidazole (**4l**). Note: This compound was additionally purified by manual column chromatography (SiO$_2$, Hexane/EtOAc (7/3)) Light-brown crystals. Yield: 0.14 mmol (44 mg, 28%), mp = 118–119 °C. *R*$_f$ 0.35 (hexane/EtOAc, 6:4). $^1$H NMR (DMSO-*d*$_6$): δ 7.88–7.86 (m, 2H); 7.59–7.52 (m, 5H); 7.04–7.02 (m, 2H); 3.80 (s, 3H); 1.38 (s, 6H) ppm. $^{13}$C {$^1$H} NMR (DMSO-*d*$_6$, APT mode): δ 161.8 (C); 161.7 (C); 160.2 (C); 135.7 (CH); 131.0 (C); 130.9 (CH); 128.8 (CH); 128.2 (CH); 118.6 (C); 115.1 (CH); 101.9 (C); 55.3 (CH$_3$); 24.2 (CH$_3$) ppm. IR (DRA): ν 2925, 2852, 1590, 1521, 1490, 1443, 1243, 1167, 1092, 1022, 983, 922, 830, 812, 777, 639, 571 cm$^{-1}$. MS calcd *m/z* 310 [M]$^+$. Found (EI): *m/z* 310 [M]$^+$. Elemental analysis calcd for: C$_{18}$H$_{18}$N$_2$OS: C, 69.65; H, 5.85; N, 9.02. Found: C, 69.67; H, 5.85; N, 9.00.

4-((2,6-Dichlorophenyl)thio)-2,2-dimethyl-5-phenyl-2*H*-imidazole (**4m**). Note: This compound was additionally purified by manual column chromatography (SiO$_2$, Hexane/EtOAc

(7/3)) Light-yellow crystals. Yield: 0.175 mmol (61 mg, 35%), mp = 132–133 °C. $R_f$ 0.35 (hexane/EtOAc, 6:4). $^1$H NMR (DMSO-$d_6$): δ 7.89 (dd, *J* = 7.7, 1.9 Hz, 2H); 7.67 (s, 1H); 7.65 (s, 1H); 7.62–7.57 (m, 3H); 7.55–7.51 (m, 1H); 1.37 (s, 6H) ppm. $^{13}$C {$^1$H} NMR (DMSO-$d_6$, APT mode): δ 161.2 (C); 158.3 (C); 139.9 (C); 132.6 (CH); 131.2 (CH); 130.6 (C); 129.2 (CH); 129.0 (CH); 128.0 (CH); 127.3 (CH); 102.3 (C); 24.1 (CH$_3$) ppm. IR (DRA): ν 2928, 1613, 1529, 1488, 1258, 1213, 1159, 1107, 1024, 980, 871, 773, 718, 692, 572 cm$^{-1}$. MS calcd *m/z* 348 [M]$^+$; 350 [M + 2]$^+$. Found (EI): *m/z* 348 [M]$^+$; 350 [M + 2]$^+$. Elemental analysis calcd for: C$_{17}$H$_{14}$Cl$_2$N$_2$S: C, 58.46; H, 4.04; N, 8.02. Found: C, 58.44; H, 4.05; N, 8.02.

## 3. Results and Discussion

Novel arylthioimidazoles were prepared by transition metal-free C-H arylthiolation of 2*H*-imidazole **1** with thiophenols **2**. This reaction can be considered as a special case of nucleophilic substitution of hydrogen (S$_N$$^H$) to be proceeded via the "addition-elimination" (S$_N$$^H$ AE) pathway, with N-oxide moiety acting as a leaving group. As a result, the desired compounds have been obtained as hydrochloride salts; the latter can be easily converted into their corresponding bases (Scheme 2).

**Scheme 2.** Transition metal-free C-H arylthiolation of 2*H*-imidazole 1-oxides **1** (red) with thiophenol **2** (blue).

To determine the optimal conditions for these couplings, a reaction between 2*H*-imidazole-1-oxide **1a** and thiophenol **2a** was chosen as the model (Scheme 3). The effect of solvent, activator, temperature, and reaction time has been investigated. For the first time, the desired compound **3a** was obtained with a yield of 15% by stirring the reaction mixture from 0 °C to ambient temperature in toluene for 6 h, followed by the addition of acetyl chloride (Table 1, Entry 1). The further iteration resulted in a yield of 56% when acetone was used as a solvent (Table 1, Entry 3). Finally, a more thorough choice of solvent, temperature, and reaction time allowed us to obtain a product with 74% yield under the following conditions: in dimethyl carbonate (DMC) from 0°C to room temperature and stirring for 6 h (Table 1, Entries 4–13). All attempts to replace acetyl chloride as an activator were found to lead to a decrease in yield (Table 1, Entry 14) or to isolated starting materials (Table 1, Entries 15–18). It should be mentioned that DMC is one of the preferable green solvents for synthesis due to its low toxicity, biodegradability, and absence of irritable and mutagenic effects [39].

**Scheme 3.** Model reaction for optimization of C-H arylthiolation of 2*H*-imidazole 1-oxide **1a** (red) with thiophenol **2a** (blue).

**Table 1.** Optimization of the C-H arylthiolation of 2*H*-imidazole 1-oxide **1a** with thiophenol **2a** (bold for the best result of optimization).

| Entry [a] | Solvent | Activator (1 Equiv) | Temperature (°C) | Time (h) | Yield (%) |
|---|---|---|---|---|---|
| 1 | Toluene | AcCl | 0 to rt | 6 | 15 [b] |
| 2 | EtOAc | AcCl | 0 to rt | 6 | 24 [b] |
| 3 | Acetone | AcCl | 0 to rt | 6 | 56 [b] |
| 4 | Hexachloroacetone/acetone (4/1) | AcCl | 0 to rt | 6 | 54 [b] |
| 5 | Chlorobenzene/acetone 4/1 | AcCl | 0 to rt | 6 | 40 [b] |
| **6** | **DMC** | **AcCl** | **0 to rt** | **6** | **74 [b]** |
| 7 | DMC | AcCl | 0 to 50 | 6 | 40 [b] |
| 8 | DMC | AcCl | rt | 6 | 55 [b] |
| 9 | DMC | AcCl | 0 to rt | 2 | 36 [b] |
| 10 | DMC | AcCl | 0 to rt | 3 | 45 [b] |
| 11 | DMC | AcCl | 0 to rt | 4 | 58 [b] |
| 12 | DMC | AcCl | 0 to rt | 5 | 64 [b] |
| 13 | DMC | AcCl | 0 to rt | 7 | 74 [b] |
| 14 | DMC | Trichloroacetyl chloride | 0 to rt | 6 | 42 [b] |
| 15 | DMC | BzCl | 0 to rt | 6 | 0 [c] |
| 16 | DMC | TFAA | 0 to rt | 6 | 0 [c] |
| 17 | DMC | TMS-Cl | 0 to rt | 6 | 0 [c] |
| 18 | DMC | Ac$_2$O | 0 to rt | 6 | 0 [c] |

[a] All reactions were carried out using 1 mmol of each substrate. [b] Isolated yield. [c] Starting materials were recovered.

In our previously reported studies of the $S_N^H$ methodology, we investigated reactions of 2*H*-imidazole 1-oxides **1** with various substrates of aromatic and heteroaromatic nature [8–10]. The current work deals with the formation of C-S bonds in contrast to published C-C couplings. The S-nucleophilicity of thiophenols is obviously much higher than the C-nucleophilic properties of carbon centers of pyrroles, indoles, and phenols. Thereby, this reactivity feature affects the regioselectivity for the studied reaction and thus results exclusively in the C-S coupling products. It is also worth noting that the analogues reactions with polyphenols did not lead to C-O bond formation products following the C-S coupling logic. This observation could account for the greater electronegativity (lower nucleophilicity) of the oxygen atom compared with sulfur.

Finally, we have managed to obtain 11 arylthio-2*H*-imidazoles **3a-k** as hydrochloride salts in yields of 42–90%, as well as four compounds **4a, 4i, 4l, 4m** as bases in 28–90% yields (Scheme 4). Pure compounds **4l** and **4m** in the forms of hydrochloride have not been able to be isolated solely and thus require further extra purification by column chromatography.

According to the plausible reaction mechanism (Scheme 5), at the first stage, acetyl chloride is attached to the N-oxide group of 2*H*-imidazole 1-oxide **1** to obtain a structure **1.1**, which is equal to **1.2** with a positive charge on the C(5) atom. This form is likely to undergo a nucleophilic attack from the active S-H bond of thiophenol **2** with the formation of intermediate **1.3** to be stabilized by the positive charge on the sulfur atom by the chloride anion. As a result of the acetic acid elimination, a new C-S bond is formed, with sulfenylated imidazole derivatives in the form of hydrochloride **3** being formed.

**Scheme 4.** The developed arylthio-2*H*-imidazoles (red color indicates the imidazole ring; blue color shows the thioaryl moiety; * yield of two steps).

**Scheme 5.** Plausible mechanism for C-H arylthiolation of 2*H*-imidazole 1-oxide **1** (red) with thiophenols **2** (blue).

## 4. Conclusions

In summary, 15 novel arylthioimidazoles of various architectures, including water-soluble hydrochloride forms, were synthesized in yields of up to 90%. In particular, the strategy of nucleophilic substitution of hydrogen ($S_N^H$) was first applied in reactions of 2*H*-imidazole-1-oxides with thiophenols. The elaborated synthetic method demonstrated a high level of regioselectivity, thus providing only C-S coupling products in the absence of C-C coupling by-products. The synthesized arylthiolated 2*H*-imidazoles could be considered challenging molecules in the field of medicinal chemistry and advanced materials, as well as valuable intermediates for further chemical modifications.

**Supplementary Materials:** The following supporting information can be downloaded at: https://www.mdpi.com/article/10.3390/chemistry5030100/s1, Figure S1–S31: Copies of NMR spectra for **3a-k**, **4a,i,l,m**.

**Author Contributions:** Conceptualization, O.N.C., V.N.C., and M.V.V.; methodology, E.A.N. and T.D.M.; investigation, E.A.N., N.F.V., and T.D.M.; writing—original draft preparation, E.A.N. and T.D.M.; writing—review and editing, M.V.V. and V.N.C.; visualization, N.F.V.; supervision, M.V.V.; project administration, V.N.C. and O.N.C. All authors have read and agreed to the published version of the manuscript.

**Funding:** This work was financially supported by the Russian Science Foundation (RSF), project № 23-63-10011.

**Data Availability Statement:** The data presented in this study are available upon request from the corresponding author and co-authors.

**Conflicts of Interest:** The authors declare no conflict of interest.

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
