# Peer review of "Metal-Free Eliminative C-H Arylthiolation of 2H-Imidazole N-Oxides with Thiophenols"

_chemistry, doi:10.3390/chemistry5030100_

Round 1
Reviewer 1 Report
Authors described interesting method for the synthesis of 4-arylthioimidazoles. This is an extension of previous work of authors (e.g JOC 2021, 86, 19, 13702) that takes advantage of the reactivity of 2H-Imidazole N-oxides with nucleophiles. I have no major complaints about this work. However some question came up after reading. Why only four derivatives were converted to free bases. Also, it is interesting how aliphatic thiols will react with 2H-Imidazole N-oxides.
In my opinion selfcitation (10 out of 35) is too high.
I suggest to correct following errors:
line 38 azaheterocycle
line 76 TLC
line 78 thiosalicylic
line 96 stirr
line 191 -108,40 (c ? - did you mean singlet ?
line 203 and 206 DMSO
line 265 thiophenol 2a (blue)
line 305 regioselectivity
Para substituted phenyl gives in 1H NMR AA'XX' system that is not equivalent to two dublets. The signals should be described as multiplets not a dublets. Anyway, two coupled protons have exactly the same coupling constants. The difference comes from low digital resolution of spectra. Check for example line 125: (3d) 1H NMR 8.07 (d, J = 8.3 Hz, 2H); 7.74 (d, J = 8.2 Hz, 2H).
Author Response
Reviewer 1
Authors described interesting method for the synthesis of 4-arylthioimidazoles. This is an extension of previous work of authors (e.g JOC 2021, 86, 19, 13702) that takes advantage of the reactivity of 2H-Imidazole N-oxides with nucleophiles. I have no major complaints about this work. However some question came up after reading.
We kindly thank the reviewer for the positive feedback on our manuscript. We have thoroughly revised the manuscript and corrected the mistakes and inaccuracies found.
Why only four derivatives were converted to free bases.
All structures can be transformed to the corresponding free bases. In this work, four derivatives with different substituents were successfully converted to demonstrate the synthetic possibilities of the elaborated method.
Also, it is interesting how aliphatic thiols will react with 2H-Imidazole N-oxides.
This work is focused on the reactivity of aromatic thiols with 2H-imidazole 1-oxides. One of the next studies are considered to be included the reactivity issues of azaheterocyclic substrates with aliphatic thiols.
In my opinion selfcitation (10 out of 35) is too high.
The self-citation number has been reduced to 6 of 39
I suggest to correct following errors:
Minor Revisions:
- line 38 azaheterocycle.
The word has been corrected.
- line 76 TLC
The abbreviation has been corrected.
- line 78 thiosalicylic
The word has been corrected
- line 96 stirr
The word has been corrected.
- line 191 -108,40 (c ? - did you mean singlet ?
The mistake has been corrected.
- line 203 and 206 DMSO
The mistake has been corrected.
- line 265 thiophenol 2a (blue)
The mistake has been corrected.
- line 305 regioselectivity
The word has been corrected.
- Para substituted phenyl gives in 1H NMR AA'XX' system that is not equivalent to two dublets. The signals should be described as multiplets not a dublets. Anyway, two coupled protons have exactly the same coupling constants. The difference comes from low digital resolution of spectra. Check for example line 125: (3d) 1H NMR 8.07 (d, J = 8.3 Hz, 2H); 7.74 (d, J = 8.2 Hz, 2H). The found errors have been corrected.
Reviewer 2 Report
In this manuscript, the authors present nucleophilic attack reaction of imidazole N-oxides with thiophenols. This research group has several other papers demonstration closely analogous transformations, with other nucleophiles. It is not particularly surprising that these transformations work, but it is sufficiently useful and interesting to warrant publication in Chemistry. I would be interested in knowing in the lower chemical yield cases (4l and 4m), the nature of any side products – is coupling through an arene ring carbon of the thiophenol see, is it just recovered SM, or is it just a messy reaction.
My biggest issue with the paper is the melting points. These are usually quote with ranges of 4 or 5 oC, which is too wide a range for a compound of good purity. In compound 3j, the elemental analysis for carbon is too far off. In the SI for the for 3k, the 13C spectrum (which is actually labelled as 3i) has an additional multiplet at ca. 21 ppm. This is neither from the compound nor CF3CO2D. Otherwise the SI is quite good.
Beyond that, there are a few typos to take care of. Dimethyl carbonate is two words throughout the manuscript.
Line 20, Abstract – thiophenols Line 41 – aryl halides Line 66 – CDCl3
Line 76 – TLC
Line 113 – it’s %C (not S) which is 67.31(%)
Line 276 – phenols
In Scheme 5, cpd 1.2, there’s an open square at the formally cationic carbon that has no place.
Line 305-306 – regioselectivity
Line 359, Ref 16. The journal abbreviation is normally Eur. J. Org. Chem.
These are included in the previous box.
Author Response
In this manuscript, the authors present nucleophilic attack reaction of imidazole N-oxides with thiophenols. This research group has several other papers demonstration closely analogous transformations, with other nucleophiles.
It is not particularly surprising that these transformations work, but it is sufficiently useful and
We kindly thank the reviewer for the positive feedback on our manuscript. We thoroughly have revised the manuscript and corrected the found mistakes and inaccuracies.
- I would be interested in knowing in the lower chemical yield cases (4l and 4m), the nature of any side products – is coupling through an arene ring carbon of the thiophenol see, is it just recovered SM, or is it just a messy reaction.
The products of C-S bond formation in cases of 4l and 4m are most likely to be partly decomposed. We tried to wash the precipitate with hexane but this procedure did not result in the desired pure compounds. Moreover, there were no any side products detected.
- My biggest issue with the paper is the melting points. These are usually quote with ranges of 4 or 5 oC, which is too wide a range for a compound of good purity.
The melting points have been remeasured.
- In compound 3j, the elemental analysis for carbon is too far off.
The elemental analysis data have been remeasured.
- In the SI for the for 3k, the 13C spectrum (which is actually labelled as 3i) has an additional multiplet at ca. 21 ppm. This is neither from the compound nor CF3CO2D. Otherwise the SI is quite good.
The 13C NMR spectrum for compound 3j was recorded in CF3COOD with the addition of CD3COOD for calibration of the spectrometer as an internal standard. The multiplet at 20 ppm is a signal of CD3COOD. The relevant corrections have been performed in the revised version of the manuscript and the ESI.
- Dimethyl carbonate is two words throughout the manuscript.
The mistake has been corrected.
- Line 20, Abstract – thiophenols
The mistake has been corrected.
- Line 41 – aryl halides
The mistake has been corrected.
- Line 66 – CDCl3
The mistake has been corrected.
- Line 76 – TLC
The mistake has been corrected.
- Line 113 – it’s %C (not S) which is 67.31(%)
The misspell has been corrected.
- Line 276 – phenols
The misspell has been corrected.
In Scheme 5, cpd 1.2, there’s an open square at the formally cationic carbon that has no place.
The scheme 5 has revised.
Line 305-306 – regioselectivity
The mistake has been corrected
Line 359, Ref 16. The journal abbreviation is normally Eur. J. Org. Chem.
The mistake has been corrected.
Reviewer 3 Report
The manuscript "Metal-Free Eliminative C-H Arylthiolation of 2H-Imidazole N-oxides with Thiophenols" reports a new approach to arylthiolated-imidazoles via a nucleophilic substitution of hydrogen in non-aromatic azaheterocyclic substrates. The approach is novel, the SI is fine. The experiments were well planned. The manuscript is suitable for Chemistry, and can be accepted after minor revision.
1. Fig. 1: Azathioprine should be labeled in a similar ways with compounds I-IV (i.e. state its bioligicl activity). In my opinion compound V is structurally excessive for this Fig., since it does not bear PhS moiety.
2. Line 38: delete Russian symbols.
3. The notes “(red color indicates the aвzaheterocycle fragment, blue color shows the sulfenyl moiety). and etc.” should be deleted everywhere.
4. In EAs calculated: elements, which were not examined (S, Hal, and etc.) , should be deleted.
5. MS (EI): m/z should be reported as calculated and found.
6. Line 113: calculated С is missing.
7. Scheme 5 is too complicated. It should be revised (arrows and arcs).
Author Response
The manuscript "Metal-Free Eliminative C-H Arylthiolation of 2H-Imidazole N-oxides with Thiophenols" reports a new approach to arylthiolated-imidazoles via a nucleophilic substitution of hydrogen in non-aromatic azaheterocyclic substrates. The approach is novel, the SI is fine. The experiments were well planned. The manuscript is suitable for Chemistry, and can be accepted after minor revision.
We kindly thank the reviewer for the positive feedback on our manuscript. We have thoroughly revised the manuscript and corrected the mistakes and inaccuracies found.
- Fig. 1: Azathioprine should be labeled in a similar ways with compounds I-IV (i.e. state its bioligicl activity). In my opinion compound V is structurally excessive for this Fig., since it does not bear PhS moiety.
In introduction section, we would like to demonstrate the diversity of imidazoles that contain arylthiol moiety in biologically active molecules. Example V is an immunosuppressive drug that includes both imidazole and heteroarylthio moieties. Thus, we suppose that this example is relevant and of particular interest to the readers of Chemistry.
- Line 38: delete Russian symbols.
The mistake has been corrected.
- The notes “(red color indicates the aвzaheterocycle fragment, blue color shows the sulfenyl moiety). and etc.” should be deleted everywhere.
This explanation is the requirement of the MDPI Editorial Office. If we delete it now, the Editorial Office can ask us to include this note at the proofreading stage.
- In EAs calculated: elements, which were not examined (S, Hal, and etc.) , should be deleted.
The uncalculated elements have been removed.
- MS (EI): m/z should be reported as calculated and found.
MS(EI) was revised and reported as calculated and found.
- Line 113: calculated Сis missing.
The missing value has been added.
- Scheme 5 is too complicated. It should be revised (arrows and arcs).
Scheme 5 has been simplified.
Reviewer 4 Report
This manuscript describes the full details of the synthesis of a variety of 2H-imidazoles having an arylthio substituent via the reaction of 2H-imidazole-1-oxide and thiophenols mediated by acetic chloride as an activator. As authors mentioned, the products of this research work would give a new choice of the seeds for medicinal chemistry and building block of new materials.
Chemistries described in this paper would be a nice piece of work and of interest for a number of researchers in this field.
I recommend this manuscript for publication in Chemistry.
A few minor points:
Page 1, Title and Abstract: The expression “metal-free” and “metal-free C-H/S-H coupling” may cause the misunderstanding for the leaders. The reaction described is an addition-elimination of cyclic imine oxides and thiols mediated by acetyl chloride. This should not be categorized as coupling reaction. Nothing activate the C-H bond nor S-H bond in the reaction descri
Page 7, Table 2: An abbreviation, DCM was given for dimethylcarbonate in entry 6.
I don not understand why DCM was not used for the entries 8 to 18.
Page 9, line 303-304: This sentence seems to be contradictory for me. Nucleophilic substitution should not be categorized C-H/S-H coupling.
Author Response
This manuscript describes the full details of the synthesis of a variety of 2H-imidazoles having an arylthio substituent via the reaction of 2H-imidazole-1-oxide and thiophenols mediated by acetic chloride as an activator. As authors mentioned, the products of this research work would give a new choice of the seeds for medicinal chemistry and building block of new materials. Chemistries described in this paper would be a nice piece of work and of interest for a number of researchers in this field.I recommend this manuscript for publication in Chemistry.
We kindly thank the reviewer for the positive feedback on our manuscript. We thoroughly revised the manuscript and corrected mistakes and inaccuracies.
A few minor points:
- Page 1, Title and Abstract: The expression “metal-free” and “metal-free C-H/S-H coupling” may cause the misunderstanding for the leaders. The reaction described is an addition-elimination of cyclic imine oxides and thiols mediated by acetyl chloride. This should not be categorized as coupling reaction. Nothing activate the C-H bond nor S-H bond in the reaction described.
The categorization “C-H/S-H” was removed and the abstract has been rewritten. The expression “metal-free” in the title is widespread and in common use:
- Metal-Free C−H Alkyl(Aryl)Thiolation and Chlorination of Electron-Rich Arene with Thiosulfinate https://dx.doi.org/10.2139/ssrn.4247696
- Transition metal free K2CO3 mediated thioarylation, selenoarylation and arylation of 2-aminomaleimides at ambient temperature https://doi.org/10.1016/j.tet.2019.130486
- Metal-free C–H thioarylation of arenes using sulfoxides: a direct, general diaryl sulfide synthesis https://doi.org/10.1039/C6CC07627K
- Metal free photoinduced C(sp3)–H thioarylation https://doi.org/10.26434/chemrxiv-2022-hfftk
Thus, we believe that expression “metal-free” will not be confused to the readers.
- Page 7, Table 2: An abbreviation, DCM was given for dimethylcarbonate in entry 6. I do not understand why DCM was not used for the entries 8 to 18.
The table has been corrected
- Page 9, line 303-304: This sentence seems to be contradictory for me. Nucleophilic substitution should not be categorized C-H/S-H coupling.
We agree with the reviewer that readers might be confused. The sentence has been rewritten in the revised version of the manuscript.